# Target Prediction by Multiple Virtual Screenings: Analyzing the SARS-CoV-2 Phenotypic Screening by the Docking Simulations Submitted to the MEDIATE Initiative

**DOI:** 10.3390/ijms25010450

**Published:** 2023-12-29

**Authors:** Silvia Gervasoni, Candida Manelfi, Sara Adobati, Carmine Talarico, Akash Deep Biswas, Alessandro Pedretti, Giulio Vistoli, Andrea R. Beccari

**Affiliations:** 1Dipartimento di Scienze Farmaceutiche, Università Degli Studi di Milano, Via Mangiagalli, 25, I-20133 Milano, Italy; silvia.gervasoni@dsf.unica.it (S.G.); sara.adobati@gmail.com (S.A.); alessandro.pedretti@unimi.it (A.P.); 2Department of Physics, Università di Cagliari, I-09042 Monserrato, Italy; 3EXSCALATE, Dompé Farmaceutici S.p.A., Via Tommaso De Amicis, 95, I-80131 Napoli, Italy; candida.manelfi@dompe.com (C.M.); carmine.talarico@dompe.com (C.T.); akashdeep.biswas@dompe.com (A.D.B.); andrea.beccari@dompe.com (A.R.B.)

**Keywords:** SARS-CoV-2, phenotyping screening, in silico target identification, multiple docking simulations, consensus strategy

## Abstract

Phenotypic screenings are usually combined with deconvolution techniques to characterize the mechanism of action for the retrieved hits. These studies can be supported by various computational analyses, although docking simulations are rarely employed. The present study aims to assess if multiple docking calculations can prove successful in target prediction. In detail, the docking simulations submitted to the MEDIATE initiative are utilized to predict the viral targets involved in the hits retrieved by a recently published cytopathic screening. Multiple docking results are combined by the EFO approach to develop target-specific consensus models. The combination of multiple docking simulations enhances the performances of the developed consensus models (average increases in EF1% value of 40% and 25% when combining three and two docking runs, respectively). These models are able to propose reliable targets for about half of the retrieved hits (31 out of 59). Thus, the study emphasizes that docking simulations might be effective in target identification and provide a convincing validation for the collaborative strategies that inspire the MEDIATE initiative. Disappointingly, cross-target and cross-program correlations suggest that common scoring functions are not specific enough for the simulated target.

## 1. Introduction

In the last years, phenotypic screening methods have gained a remarkable relevance for their capacity to identify promising hits for well-defined pathological conditions [1,2]. The main reason for such a success can be found in the recent advancements in cell-based technologies that allow the development of reliable disease models [3,4]. These cellular models can express some key clinically relevant features and allow the effects of a tested molecule to be monitored in phenotyping screening campaigns [5]. The phenotypic screening implies a paradigm shift from a target-based approach to a disease (or phenotype)-based approach. This strategy becomes particularly productive for complex or poorly understood pathologies for which target identification and validation can be pursued with difficulty [6]. Phenotypic screening can identify active molecules without unraveling the corresponding mechanism of action. While not always mandatory for the drug discovery process, understanding the targeted protein(s) can be of invaluable relevance during the hit and lead optimization phases [7].

Hence, a phenotypic screening is usually paired by a set of studies for determining the mechanism of action, at least for the most promising compounds. Such a challenging task is supported by a rich arsenal of experimental methods, including biochemical affinity analyses [8,9], “omics” profiling [10,11,12], and mutational studies [13]. Furthermore, this task can greatly benefit from computational studies, which become even more necessary when dealing with huge datasets [14].

Some computational approaches can involve in-depth analyses of the phenotype features to predict the potential mechanism of action based on cell viability or cell morphology profiles [15]. Other computational approaches exploit common ligand-based and/or cheminformatics approaches and are substantially based on similarity analyses [16,17]. By contrast, docking simulations are rarely utilized for target identification apart from focused studies to confirm the reliability of the targets proposed by other predictive approaches [18]. The limited use of docking simulations in target identification can be explained by the greater complexity of these calculations, which are markedly more time-consuming than simple similarity searches. Nevertheless, the recent advancements in high-performing computing for docking simulations as well as the availability of optimized docking engines render these simulations amenable for target prediction, at least in well-defined contexts [19]. In this scenario, one can appreciate the various methods recently proposed for the so-called inverse docking methodology, in which a single molecule is docked into a set of therapeutically relevant proteins to reveal the potential targets and then the related mechanisms of action [20,21]. For example, this approach was recently applied for the target identification of compounds active against SARS-CoV-2 [22].

On these grounds, the present study aims to assess the efficacy of multiple docking simulations to identify the proteins targeted by the hits retrieved during a phenotyping screening. Specifically, the study is based on a recently published large-scale repurposing campaign performed within an EU-funded project (H2020-EXSCALATE4COV). About 8700 compounds underwent cytopathic screening on VERO-E6 cells to reveal 110 active compounds, namely endowed with an anti-cytopathic IC_50_  <  20 µM [23].

To this end, the study utilizes the docking simulations focused on the viral targets and submitted to MEDIATE [24,25]. This collaborative computing initiative invites researchers worldwide to contribute to docking simulations by exploiting shared sets of purposely prepared and annotated protein structures and standardized ligands libraries. In detail, the results analyzed here are derived from three sets of docking simulations (as generated by PLANTS, LiGen, and GOLD). They comprise an extended set of 14 viral binding sites (from 12 viral proteins) and encompass a large portion of the experimentally screened ligands. The collected virtual screening results were employed to develop consensus models able to predict the viral targets for the retrieved hits. Along with proposing potential targets, the performed analyses provide a set of predictive models for each considered SARS-CoV-2 target. All the consensus models were generated by the EFO approach (Enrichment Factor Optimization), which generates its predictive equations by linearly combining a user-defined number of variables through an exhaustive search algorithm guided by a quality function, in which the resulting EF1% value has a prevailing role [26].

It should be noted that the approach described here was applied to SARS-CoV-2 targets as a case study, but the multiple virtual screening approach here proposed might find successful applications in all target identification studies based on phenotyping screening campaigns. Similarly, the collaborative computing activities proposed by the MEDIATE initiative is focused on SARS-CoV-2 targets but can be clearly applied for docking simulations and virtual screening campaigns in all therapeutic areas.

Finally, the availability of the docking results for the same ligands on different proteins and by diverse docking tools allowed unprecedented comparative and correlative analyses to be performed.

## 2. Results

### 2.1. Docking Simulations and the Mediate Initiative

As stated in the Introduction, the study is based on the docking simulations submitted to the MEDIATE initiative [24]. The libraries shared by MEDIATE include a dataset of about 8700 safe-in-human molecules, which roughly corresponds to those experimentally screened in the repurposing campaign [23]. Here, attention was focused on 12 viral proteins, including a total of 14 target binding sites. (Nsp-12 includes two additional allosteric pockets.) The docking simulations submitted to MEDIATE include three almost complete sets of virtual screening campaigns, as carried out by using PLANTS [27], LiGen [28], and GOLD [29]. Even though the collected results did not include all 8700 compounds for all considered binding pockets and all docking programs, a common subset of about 6000 molecules for which docking results are available for all considered binding sites and for the three docking engines was identified. Such a dataset included 59 retrieved hits and thus was well suited for virtual screening campaigns, with a percentage of active compounds equal to about 1%. As detailed under Methods, the docking results were then rescored by using Rescore+ [30], and the computed scores were used to develop consensus models by applying the EFO approach [26]. 

### 2.2. Comparison of the Performances of the Employed Docking Tools

Figure 1 shows the performances reached by the three docking programs for the 14 binding pockets, as encoded by the EF1% mean for the best 20 EFO models. The comparison of the reported mean values reveals that there is no correlation between the performances of the three docking programs.

Although they reach comparable EF1% overall means, the docking simulations show similar performances only for two targets (i.e., N-Prot and Pl-Pro). In most cases, a program outperforms, compared to the other two. This situation is particularly remarkable for LiGen with Nsp14, for PLANTS with Nsp9, and for GOLD with Nsp16. The different performances of the three docking engines suggest that they do not depend on an intrinsic complexity of the targets but are mostly related to the implemented docking algorithms. 

Appendix A details the performances of the three sets of VS campaigns, reporting the EF1% value for the best models and the number of active molecules in their top 1%. The best EF1% values show mean performances that are even more superimposable. Similarly, the number of actives ranges between 3 and 7, with a mean value around 4.5 in all the three sets of docking runs. 

The comparison of hits retrieved within the top 1% by all performed simulations reveals that there is a significant occurrence of frequent hits, namely of ligands, that are predicted as actives (namely found within the top 1%) for several targets (up to six different binding sites). In detail, the occurrence of unique hits, namely ligands that are predicted to be active for only one target, is lower for LiGen, compared to PLANTS and GOLD. In contrast, the frequencies of ligands that are active at most on two binding sites are rather similar in all docking runs and around 50%. This means that all programs provide rather specific suggestions concerning the possible mechanism(s) of action for about one half of the retrieved hits. The frequencies of ligands that are predicted as actives on many targets (i.e., >4) are similar in all docking programs and around 25%. The interpretations of these frequent hits are either that they elicit their antiviral effect through a polypharmacological profile or that the interacting capacity of some large and/or polar ligands tends to be overestimated by all docking programs (and by all computed scores). The obtained results indicate that the three docking tools are similarly affected by this possible bias. 

The analysis of common retrieved hits between the three docking engines reveals similar results. Indeed, the three possible pairs of docking programs always share 11 common ligands, and only three molecules are predicted as actives on the same target by all programs. The shared molecules include salinomycin, which is predicted to be active on Nsp13, Nsp12_ortho, and Nsp14, while loperamide is predicted as active on Nsp3, and hypericin is seen as a potential allosteric modulator for Nsp12. 

### 2.3. Consensus Analyses and Predicted Targets

The MEDIATE initiative is based on the concept that proper combinations of different docking results should lead to enhanced predictive powers. Hence, the results of the three sets of docking runs were combined by using the EFO algorithm. In other words, for each target, the computed scores from two or three docking simulations were merged to generate the corresponding consensus EFO-based equations.

Figure 2 shows the performances derived by these consensus analyses as encoded by their EF1% mean values. In detail, Figure 2 compares the performances reached by combining the scores (1) from all three docking programs, (2) from the three possible pairs of docking tools, and (3) the average performances of the three docking runs taken alone. 

Figure 2 shows that the combination of the three sets of docking scores enhances the resulting performances in all considered targets. In detail, the enhancement is greater than 40% in 6 cases out of 14, with an enhancement mean equal to 34.4%. The effect is particularly relevant for 3CL-Pro, Nsp14, and Nsp16, where the consensus approach leads to a doubling of the EF1% values. Such an enhancement is reflected in the number of hits retrieved in the top 1%, ranging from 5 to 9 molecules, with an average equal to 6.2. Similar enhancements are seen when analyzing the best EF1% values (Appendix A). The average increase in the best EF1% values is equal to 36.6%, with the remarkable case of Nsp14 showing an EF1% increase of 71.5%.

The analysis of the predictive results reached by combining the three possible pairs of docking runs reveals rather similar performances with the GOLD–PLANTS pair, which provides slightly better results, compared to the other two combinations. The EF1% means are equal to 6.5, 6.2, and 6.0 for GOLD–PLANTS, GOLD–LiGen, and LiGen–PLANTS, respectively. As seen previously, there is no correlation between the performances of the three pairs of docking results, a finding that depends on the differences already seen for the performances of each docking simulation. Notably, the combinations of the pairs of docking runs induce an average increase in the performances equal to 25%.

Appendix A also includes the best consensus models generated by combining all three docking runs. A bird’s eye view of these equations reveals that the three sets of docking simulations similarly contribute to the compiled equations, although the scores from GOLD simulations are slightly less frequent. (The models comprise 16 scores from LiGen, 15 from PLANTS, and 11 from GOLD runs.) This outcome witnesses the efficacy of combining different docking simulations to optimize the resulting predictive performance. In detail, the included scores highlight the remarkable role of ionic contacts, since 10 out 14 models include at least one scoring function accounting for polar interactions. Finally, the consensus equations emphasize the relevance of PLANTS-based scores and XScore components (10 and 9 occurrences, respectively), which appear particularly effective in rescoring analyses.

Taken together, these results emphasize the relevance of combining the results from various docking simulations and reveal that even the simplest combination of two docking runs has an encouraging effect that conceivably increases when combining all docking campaigns. These results represent a compelling confirmation of the concept inspiring the MEDIATE initiative and indicate that predictive performances should be further enhanced by combining a higher number of docking simulations. These results also confirm the reliability of the EFO approach for developing consensus models by combining various socking scores. This finding is in line with recent studies that evidenced how machine learning methods can provide consensus models with very remarkable performances, even when applied to a single docking run (e.g., see Ref. [31]).

Table 1 compiles the hits retrieved in the top 1% for the 14 explored binding sites. Overall, the consensus models were able to identify only about half of the retrieved hits (31 out of 59). On one hand, this result might be ascribed to the above-mentioned underestimation of the interaction of small and hydrophobic compounds. On the other hand, this finding is explainable by considering the recently identified viral binding sites for which docking results are not yet available within the MEDIATE resources (e.g., the allosteric sites of 3CL-Pro).

In detail, the molecules identified as active (i.e., found in the top 1%) on a single target represent the most frequent case (9 out of 31, i.e., 29%). As previously seen, molecules active on at most two targets constitute 50%, and those predicted as active on more than three targets represent 26%. These percentages are very similar to those obtained by the single docking runs, thus suggesting that the combination of docking simulations does influence the occurrence of frequent hits.

While avoiding a systematic analysis of the compounds listed in Table 1, some retrieved hits deserve special attention. Among the frequent hits, the polypharmacological profile of clofazimine was experimentally investigated by Yang and coworkers, who demonstrated that it elicits antiviral activity by targeting several steps in SARS-CoV-2 replication [32]. Although their marked polarity and molecular size suggest that the remarkable ranking exhibited by 7-aminoalkoxy-quinazolines (UNC-646 and UNC-642) for several targets might be affected by the above-mentioned bias, their activity as potent inhibitors of the lysine methyltransferase G9a might indicate that they can also inhibit viral enzymes catalyzing similar transferase reactions [33]. The broad spectrum of amodiaquine was confirmed in a recent study that investigated the activity of a set of anti-malaria quinolines that interfere with the viral entry process at a post-attachment stage [34].

Even though specific antiviral activity data are available for a few compounds and on a limited number of viral targets, some predicted targets have found an experimental confirmation. The predicted activity of hypericin on 3CL-Pro was experimentally confirmed by a recent study that evidenced that this molecule is a pan-anti-α-CoV able to exert general antiviral activity against their replication [35]. Masitinib was also confirmed for its activity on Spike-ACE2 binding in two different assays [36]. Additionally, salinomycin was reported as a modest inhibitor of the spike protein [37]. 

To further assess the occurrence of frequent hits, the analysis was extended to the best 100 molecules (regardless of their activity) according to the EFO rankings for the 14 targets. Appendix A shows the common compounds between all the possible pairs of targets, while the diagonal values (in bold) report the unique hits for each target. 

The first relevant consideration is that the unique ligands represent a vast majority, and only for two pairs of targets is the percentage of shared compounds slightly higher than 10%. From a methodological point of view, the relative scarcity of shared compounds indicates that the consensus models developed here provide rather specific and target-dependent results, at least for the best-ranked molecules. This finding is particularly relevant when considering the above-mentioned docking approximations and the cross-target and cross-docking correlations discussed below and suggests that the EFO-based consensus models might have a beneficial effect for these downsides. These results also indicate that there are neither highly promiscuous binding sites nor particularly similar pairs of targets, apart from a few cases (PLPro–Nsp12_ortho and Nsp6–Nsp13 are the only pairs with more than 20 common ligands).

### 2.4. Comparative Analyses of the Computed Scores

The availability of docking results and score values for the same ligands on 14 binding sites from three different docking engines enable meaningful analyses by variously comparing the corresponding scores. Figure 3 shows the normalized mean values as computed for the 14 targets by averaging some relevant docking scores for all the simulated compounds and for the three docking programs. For the sake of completeness, Appendix A lists the (not normalized) average values for all the computed scoring functions. 

While showing conceivable differences, the score averages reported in Figure 3 reveal some similar trends. For almost all monitored scores, Nsp3 and spike targets reveal the best scores, while Nsp9 shows the worst averages. Also, 3CL-Pro, Nsp14, and Nsp16 exhibit, on average, rather good score values, while Nsp6, Nsp12 (all sites), and Nsp13 exhibit modest score means. There is an encouraging agreement between the score trends and the binding pocket characteristics previously reported [24]. On average, the pockets showing the best score averages are characterized by either large pockets (as in the case of spike, 3CL-Pro, and Nsp14) or remarkable interaction capacities (as in the case of Nsp16 and Nsp3). In contrast, small or superficial pockets with reduced interacting features yield poor score values (as in the case of Nsp9 and, to a minor extent, PL-Pro).

Appendix A compiles the correlations between the score averages for all the considered scoring functions and for the three docking programs. The first consideration is that the overall average correlation is rather low (r^2^ = 0.22). In detail, the few highly correlated pairs of scoring functions involve the three PLANTS-based scores, which indeed share parts of the algorithm to calculate them. These high cross-correlations are reflected into the averages per scoring function. There are only 8 out of 24 scoring functions with an r^2^ mean greater than 0.3, and almost all involve PLANTS-based scores. Even though there is not necessarily a direct relation between the score averages and the score values for each compound, the modest average r^2^ value emphasizes that the computed scoring functions account for the diverse characteristics of the molecular recognition process, and thus, they can be included in the same predictive models (as performed by the EFO approach).

### 2.5. Cross-Target and Cross-Docking Correlations

Along with the comparative analyses discussed above, the computed scores can also be utilized for correlative studies. Here, two kinds of possible correlations are considered: the cross-target and the cross-docking correlations. For each docking program, the former correlates the docking scores computed for the simulated ligands between the 14 targets and reveals how much the score values depend on the target and how much they can be related to the ligand features. Stated differently, low correlations indicate high signal-to-noise ratios, namely docking scores (and docking programs) able to properly parameterize the specific ligand’s interactions within the binding pockets. In contrast, high correlations are suggestive of poor signal-to-noise ratios, namely docking scores (and docking programs) that are heavily influenced by the ligand properties and are less specific for the simulated target. For each target, the cross-docking correlations compare the results as computed by the three docking programs. They allow a degree of similarity (at least in terms of scoring functions) between different docking tools to be evaluated.

Concerning the cross-target correlations, Table 2 summarizes the r^2^ averages for some relevant scores, while Appendix A comprises a set of upper triangular matrices reporting the specific r^2^ values for the 91 possible pairs of targets. Table 2 comprises both primary scores and some relevant scoring functions from rescoring analyses. The compiled r^2^ averages show the differences between scoring functions and docking engines, which allow for some relevant considerations. The first observation is that LiGen reveals an overall better specificity, compared to PLANTS and GOLD, which show similar signal-to-noise ratios. This higher specificity is also evident when considering the simple primary scores. Next, Table 2 highlights that the number of contacts possesses the highest specificity, followed by the primary scores. In contrast, XScore and the even worse MLPIns appear to be poorly dependent on the considered binding pocket.

In general, Table 2 suggests that scoring functions computed by additive algorithms unavoidably depend on the ligand features, thus showing high cross-target correlations. This appears particularly evident for scoring functions that are based on pair potentials, such as MLPIns. In contrast, scoring functions that do not involve ligand features show reduced correlations, as in the case of the number of contacts. Although there is not a clear agreement between cross-target correlations and resulting predictive powers, this kind of analysis could reveal which docking scores are able to specifically encode for the molecular recognition process. As a matter of fact, the overall correlation mean reported in Table 2 indicates that almost half the score values do not depend on the simulated target.

The observed differences between the scoring functions also influence the cross-program correlations, as reported in Figure 4 and Appendix A. Thus, MLPIns shows the highest r^2^ averages, while the number of contacts and ChemPLP show the lowest values. The better signal-to-noise ratio of the LiGen software v1.4.1 influences the cross-program correlations, and, indeed, the pairs GOLD–LiGen and PLANTS–LiGen reveal lower and similar r^2^ mean values, compared to the GOLD–PLANTS combination. Taken together, Figure 4 suggests that the docking results generated by PLANTS and GOLD possess a higher degree of similarity, compared to those produced by LiGen. This kind of analysis could be useful when combining different docking runs to avoid consensus procedures involving highly redundant results.

## 3. Materials and Methods

### 3.1. Docking Simulations

For all the performed docking simulations, the protein and ligand structures were retrieved from the shared MEDIATE resources. In detail, the set-up of the protein structures and the annotations of their binding sites were described elsewhere [24]. The ligands were prepared as described in [24], even though the tautomers were not considered for simplicity. In all performed docking simulations, one pose per ligand was generated by focusing the search within the spheres that were previously defined [24]. Docking simulations by PLANTS and GOLD were performed using the ChemPLP scoring function, as well as the speed = 1 and the virtual screening accuracy methods, respectively. The geometrical docking procedure implemented in LiGen followed a workflow based on three docking scores: the Pacman score (PS) to estimate the geometric fitting between a ligand conformation and the pocket; the chemical score (CS), representing the ligand binding energy; and the optimized chemical score (Csopt), which accounts for the ligand binding after a minimization algorithm that treats the docked ligand as a rigid body inside the binding site. 

### 3.2. Predictive Analyses

The collected docking results from the MEDIATE initiative (1 pose per ligand) were then rescored by using Rescore+, as implemented in the WarpEngine architecture [30]. Briefly, Rescore+ calculates a set of scoring functions that can be roughly subdivided into three groups: (a) interaction-specific scores (mostly calculated by the VEGA program [38]), such as APBS for ionic interactions [39], MLPInS for hydrophobic contacts, and the CHARMM-based Lennard-Jones component for non-polar interactions; (b) PLANTS-derived scores (i.e., PLP, PLP95, and ChemPLP, plus their normalized values); and (c) XScore functions (i.e., binding energy and its components) [40]. For each docking run, the final score dataset included the primary scores plus the so-computed rescoring values.

On these grounds, for each binding site and for each docking engine, predictive models were developed by the EFO approach, which was also utilized to combine the scores from multiple docking runs. Further details concerning the EFO approach can be found elsewhere [26]. In all performed analyses, the 59 included confirmed hits were labeled as actives without exception. All developed EFO equations included three variables as selected by an exhaustive search algorithm. The predictive power of the final consensus models was assessed by randomly subdividing the dataset into training (70%) and test (30%) sets and by repeating this validation 5 times to minimize the randomness.

## 4. Conclusions

The study aims to assess the possibility of exploiting multiple virtual screening campaigns to perform an in silico target deconvolution in phenotypic screening. Specifically, the study is based on a recently published cytopathic screening study for SARS-CoV-2 and exploits the docking simulations submitted to the MEDIATE initiative. The collected docking results were used to develop consensus models, which were able to predict plausible targets for 31 (out of 59) active molecules. Among them, seven proposed targets found an experimental confirmation. Overall, these results might represent an encouraging validation for the efficacy of the computational approach for target deconvolution proposed here. Despite the discussed downsides of the docking simulations, the obtained results appear to be satisfactory when considering the scarcity of hitherto published biological data involving specific SARS-CoV-2 targets and when that the analyzed simulations were focused on the viral targets only. Notably, the predicted targets should be useful for prioritizing the ongoing biochemical assays for the retrieved hits.

Altogether, the reported docking analyses provide an encouraging confirmation to the rationale for the MEDIATE initiative, emphasizing that a proper combination of the results from different docking simulations can improve the resulting predictive performances. In detail, the consensus analyses reported here reached satisfactory performances, allowing the development of predictive models with a best EF1% average equal to 10.63 and an increased average, compared to the single docking run, equal to 40%. The developed consensus models (see Appendix A) can be used to analyze docking results for those viral targets for which experimentally tested inhibitors are not yet available.

Moreover, the availability of homogeneous docking simulations involving the same ligands dataset on 14 different targets by three diverse docking programs enabled unprecedented comparative analyses to investigate the extent and the meaning of the correlations between docking scores. Thus, the cross-target correlations revealed significant differences between the computed scores and the utilized programs. On average, the reported correlations suggest that the signal-to-noise ratio of most docking scores is not satisfactory enough, since a non-negligible component of the score value is independent of the simulated target. Also, the cross-program correlations might be exploited to properly evaluate the similarity between the computed results from different docking runs. Finally, these comparative analyses appear to be particularly relevant when applying consensus strategies to extract the most informative and specific docking scores, as well as to combine non-redundant docking results.

## Figures and Tables

**Figure 1 ijms-25-00450-f001:**
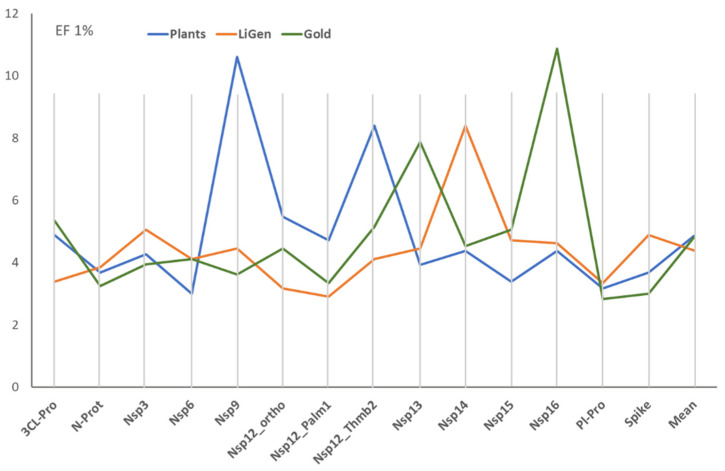
EF1% means as obtained for the 14 binding pockets by the three docking programs (The values were calculated by averaging the EF1% values for the best 20 EFO models).

**Figure 2 ijms-25-00450-f002:**
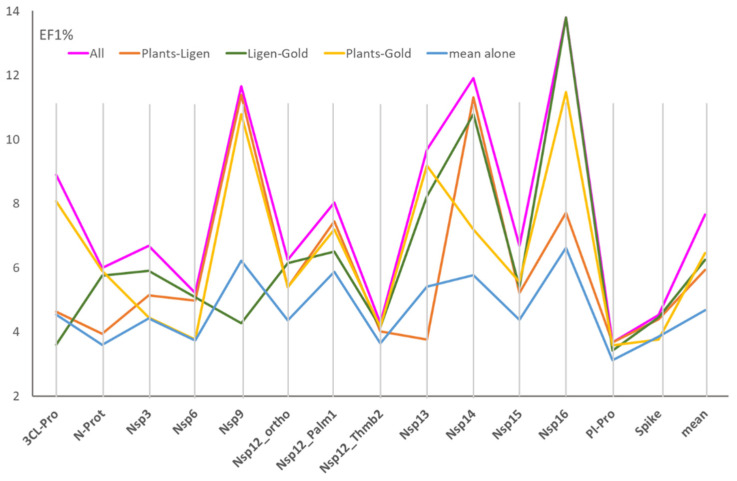
Effect of the combination of the docking scores from two or three simulations on the EF1%, compared to the mean performances of the three docking engines taken alone. (The values were calculated by averaging the EF1% values for the best 20 EFO models).

**Figure 3 ijms-25-00450-f003:**
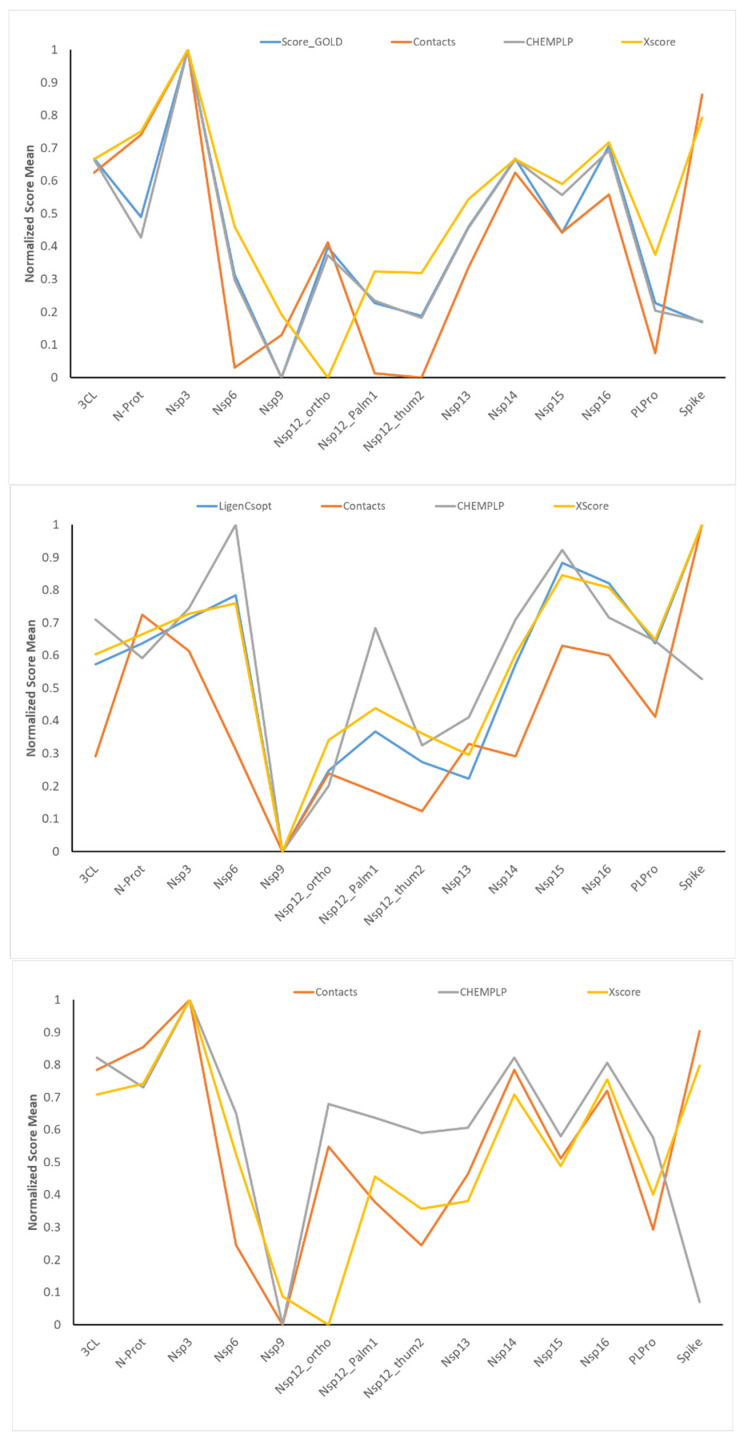
Trends of some normalized score means as computed by averaging the score values for all the simulated ligands. The top, middle, and bottom plots refer to the results from GOLD, LiGen, and PLANTS, respectively. The value 1 corresponds to the best normalized score value. The bottom panel includes only three docking scores because ChemPLP is also the primary score for PLANTS.

**Figure 4 ijms-25-00450-f004:**
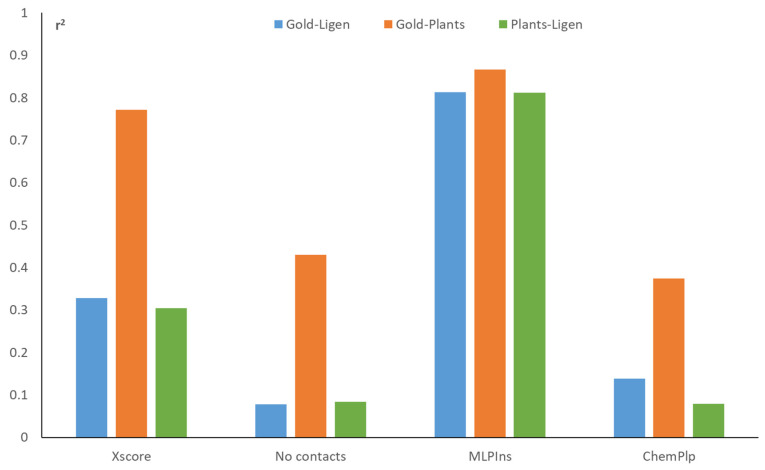
Cross-target correlation averages for some representative docking scores and for the three possible pairs of docking programs.

**Table 1 ijms-25-00450-t001:** Hits retrieved in the top 1% for the 14 explored binding pockets. For each target, the compounds are listed according to the EFO ranking. The frequent hits with more than 4 targets are evidenced by a background with a specific color code, while the unique ligands are in bold. Due to their structural similarity, UNC-0646 and UNC-0642 are considered as a sole entity.

Target	Retrieved Hits in the Top 1%						
3CL-Pro	Berzosertib	Clofazimine	Hypericin	**Ivacaftor**	KG-5	**Triclo carban**	
N-Prot	UNC-0642	Amodia quine	Clofazimine	**Halo fantrine**	PHA-665752	tetrandine	UNC-0646
Nsp3	3′-Fluoro benzyl spiperone	Loperamide	Hypericin	UK-356618	Vx-11e	WAY-600	
Nsp6	**MCOPPB**	Salinomycin	Tandutinib	UNC-0642	UNC-0646		
Nsp9	**BS-181**	CGP-71683	KG-5	UNC-0642	UNC-0646	Vx-11e	YM201636
Nsp12_ortho	CGP-71683	Clofazimine	ELN-441958	NVP-BHG712	Salinomycin	UK-356618	
Nsp12_Palm1	CGP-71683	Clofazimine	ELN-441958	Hypericin	UK-356618	Vx-11e	
Nsp12_Thmb2	ELN-441958	Loperamide	Tandutinib	tetrandine	UNC-0646		
Nsp13	CGP-71683	ELN-441958	Nelfinavir	UNC-0642	UNC-0646	VU-0364739	
Nsp14	UNC-0646	3′-Fluoro benzylspiperone	CGP-71683	ELN-441958	**GSK2194069**	Nelfinavir	Salino mycin
							+ WAY-600
Nsp15	CGP-71683	Clofazimine	Hypericin	**IPAG**	PHA-665752	tetrandine	
Nsp16	CGP-71683	3′-Fluoro benzy lspiperone	Amodia quine	Berzosertib	Clofazimine	Hypericin	VU-0364739 + YM201636
Pl-Pro	**CP-640186**	NVP-BHG712	PHA-665752	Salino mycin	YM201636		
Spike	ELN-441958	NVP-BHG712	PHA-665752	Salino mycin	**Masitinib**		

**Table 2 ijms-25-00450-t002:** Cross-target correlation (r^2^) averages for some representative scoring functions from the three utilized docking tools. The primary scores correspond to ChemPLP, CSopt, and GOLD_Score for PLANTS, LiGen, and GOLD, respectively. Notice that ChemPLP was also utilized to rescore the computed poses by LiGen and GOLD.

Score	PLANTS	LiGen	GOLD	Mean
Primary scores	0.38	0.26	0.32	0.32
XScore	0.71	0.29	0.73	0.57
No. contacts	0.30	0.03	0.32	0.22
MLPIns	0.87	0.80	0.84	0.84
ChemPLP	0.38	0.13	0.37	0.31
Mean	0.57	0.31	0.52	**0.46**

## Data Availability

All relevant data are included in the manuscript.

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
