# Peer review of "Target Prediction by Multiple Virtual Screenings: Analyzing the SARS-CoV-2 Phenotypic Screening by the Docking Simulations Submitted to the MEDIATE Initiative"

_ijms, 2023, doi:10.3390/ijms25010450_

Round 1
Reviewer 1 Report
Comments and Suggestions for Authors
The manuscript entitled "Target prediction by multiple virtual screenings: the case of the SARS-CoV-2 phenotypic screening as analyzed by the docking simulations submitted to the MEDIATE initiative" is a well conducted research in the drug repurposing field. The idea of use curated-data from different molecular docking sources can help to enhance the identification of phenotypes (properties) for target identification.
Despite the text is well-conducted, there are some minor issues to tackle before accepting the manuscript in the current form.
1)Several [Error! Bookmark not defined.]were in the body text.
2)The relevance of the MEDIATE initiative data must be clearly addressed.
3)Since any of the probed molecular docking programs enhanced the EF1% performance, what can be done to improve this value?
4)The sentence "In other words, the same strategy previously utilized to combine the computed scores from a single docking run was here utilized to combine the scores from two or three docking simulations for each target" is not clear to me. Can you elaborate a bit further?
5)The discussion about significance of Table 1 results " molecules active on at most two targets constitute the 50% and those predicted as active on more than 3 targets represent the 26%.These percentages are very similar to those obtained by the single docking runs thus suggesting that the combination of docking simulations does influence the occurrence of frequent hits". This sentence is interesting, since performance of machine learning techniques can enhance docking results, even on single run docking (Ricci-Lopez, 2021 https://doi.org/10.1021/acs.jcim.1c00511). This information most be discussed from the light of the already reported methods in molecular docking.
Comments on the Quality of English LanguageThe quality of English is acceptable, but can be improved.
Author Response
The manuscript entitled "Target prediction by multiple virtual screenings: the case of the SARS-CoV-2 phenotypic screening as analyzed by the docking simulations submitted to the MEDIATE initiative" is a well conducted research in the drug repurposing field. The idea of use curated-data from different molecular docking sources can help to enhance the identification of phenotypes (properties) for target identification.
Despite the text is well-conducted, there are some minor issues to tackle before accepting the manuscript in the current form.
1)Several [Error! Bookmark not defined.]were in the body text.
The problems in the references were fixed
2)The relevance of the MEDIATE initiative data must be clearly addressed.
The relevance and the general applicability of the MEDIATE initiative was discussed in the Introduction
3)Since any of the probed molecular docking programs enhanced the EF1% performance, what can be done to improve this value?
As stated in the Results, the predictive performances can be enhanced by combining a higher number of docking simulations. This should be allowed by exploiting new calculations submitted to the MEDIATE initiative.
4)The sentence "In other words, the same strategy previously utilized to combine the computed scores from a single docking run was here utilized to combine the scores from two or three docking simulations for each target" is not clear to me. Can you elaborate a bit further?
The sentence was rewritten to enhance its clarity
5)The discussion about significance of Table 1 results " molecules active on at most two targets constitute the 50% and those predicted as active on more than 3 targets represent the 26%.These percentages are very similar to those obtained by the single docking runs thus suggesting that the combination of docking simulations does influence the occurrence of frequent hits". This sentence is interesting, since performance of machine learning techniques can enhance docking results, even on single run docking (Ricci-Lopez, 2021 https://doi.org/10.1021/acs.jcim.1c00511). This information most be discussed from the light of the already reported methods in molecular docking.
The performances of the EFO approach was discussed from the light of the already reported methods in molecular docking.
The quality of English is acceptable, but can be improved.
The manuscript was carefully checked to improve the quality of English.
Reviewer 2 Report
Comments and Suggestions for Authors
Please see the attached file

The quality of the English language is good and the presentation is sound. I just suggest some minor phrasing changes (see above).
Author Response
More general comments:
- Although not above the limits, the title of the paper appears too long and less explicative. I suggest shortening it.
The title was a bit shortened as suggested.
- Although correctly and adequately referenced, I suggest that the Authors briefly introduce the EFO approach as done for the EXSCALATE4COV and MEDIATE initiatives in the introduction. This would drastically improve readability by smoothing the comprehension of the used approach.
The general description of the EFO approach was moved from the Methods to the Introduction.
- Page 2, “…By contrast, docking simulations are rarely utilized for target identification apart from focused studies to confirm the reliability of the targets proposed by other predictive approaches. [17] The limited use of docking simulations in target identification can be explained by the greater complexity of these calculations which are markedly more time consuming than simple similarity searches…”. In the introduction, it is suggested to briefly refer to the concept of Inverse Docking when enumerating the different approaches used to backtrace phenotypic results to drug/target mechanisms. I believe this would improve the readability and intuitively stress the need to provide the results of phenotypic screening results with mechanistic interpretations.
The concept of inverse docking was discussed in the Introduction and a case study focused on Sars-CoV-2 was mentioned.
- I believe that small adjustments in the text could guarantee a better framing of the scope of the paper. I understand that the data from the MEDIATE initiative refer to SARS-CoV-2, but the scope of the article prescinds from the target and such data are used to demonstrate the utility of a methodological approach (with even broader applications). I suggest separating the two concepts in the introduction and explicitly state in the introduction that SARS-CoV-2 is taken as case study. Furthermore, I suggest to be more cautious in the conclusions (could, might..), however explicating that the findings can are to be extended on further targets.
A paragraph was added in the Introduction to underline that the proposed approach is focused on SARS-CoV-2 data but can have a general applicability to elucidate the phenotyping screening results. The same concept was also extended to the collaborative computing supported by the MEDIATE initiative. Also here, the project was focused on SARS-CoV-2 simulations but can be applied to all therapeutic areas.
Specific comments
- I suggest that the paragraph in the conclusions (pp 14), [Moreover, the availability of homogeneous…to combine non-redundant docking results.] should be better formulated. Specifically, I suggest to: 1) change “..of most docking score is not so satisfactory” in “most docking scores”, and “not satisfactory enough”. 2) change “not-negligible” in “non-negligible” (more formal).
Modified as suggested.
- Page 2, “These cellular models are able to express all clinically relevant features and allow the efficacy of a tested molecule to be evaluated by its effects on the cellular phenotype.” Although I agree with the authors about the usefulness of cell-based model (and phenotyping screening) to recapitulate/monitor disease features through cellular markers, it is too much of a strong statement to suggest that they recapitulate ALL clinically relevant features. I suggest smoothening this statement by saying that cell-based models recapitulate some key features of pathologies and that that such cell markers are useful to monitor the effect of molecules in phenotypic screenings.
Modified as suggested.
- Page 2, “About 8700 compounds underwent cytopathic screening on VERO-E6 cells to reveal 110 compounds with an anti-cytopathic IC50 < 20 μM”. Although correct, I suggest to explicit that the 20 uM IC50 threshold was chosen to discriminate active compounds to improve the readability. • Page 3, “Among the libraries shared by MEDIATE, a dataset of about 8700 safe in man molecules roughly corresponds to those experimentally screened in the repurposing campaign [19].” I suggest to substitute the word “safe in man” with “safe-in-human”, and generally better formulate this sentence to improve readability.
Modified as suggested and the sentence was rewritten to enhance its clarity.
- Page 3 “…(Nsp-12 includes two allosteric pockets). [Error! Bookmark not defined.]”. I suggest to formulate: “Nsp-12 includes two additional allosteric pockets”, and introduce a reference/cancel the citation field.
Modified as suggested and introduced the correct reference.
- Page 4, Figure 1 and all other XY plots. I suggest introducing light vertical lines to help readability of the plot. Moreover, I warmly suggest the introduction of Y axis labels for the same reason.
Figures 1 and 2 were modified as suggested.
Page 4, “…On one hand, some of these frequent hits can elicit their antiviral effect through a polypharmacological profile and as such they can..” Since two quasi alternative interpretations are presented, I suggest being more possibilistic, e.g., “Interpretations of this results are either that…or that…”
The sentence was rewritten as suggested.
- Page 6, legend of Table 1. Making it explicit that each hit associated with > 4 targets has an unique color code would improve readability.
The legend was modified as requested.
- Page 7, “….Overall, the consensus models were able to identify only half of the retrieved hits…”. Reformulate with “Overall, the consensus models were able to identify only about half of the retrieved hits”.
Done.
- Page 7, “…In detail, the molecules identified as active on a single target represent the most frequent case (9 out of 31, i.e. 29%). As previously seen, molecules active on at most two targets constitute the 50% and those predicted as active on more than 3 targets represent the 26%. These percentages are very similar to those obtained by the single docking runs thus suggesting that the combination of docking simulations does influence the occurrence of frequent hits..”. In this sentence it would be beneficial to briefly reminder to the reader which criterion was used to identify a compound as active by docking.
The criterion (namely within the Top 1%) was reminded in the text.
- Page 8, “…apart from few cases.” I believe that here explicitly stating the cases would improve the clarity.
The two pairs of proteins sharing more than 20 common binders were mentioned.
- Page 12, “Taken together, Figure 6 suggests..”, probably the Authors meant “Figure 4”.
Corrected as suggested.
- Page 12, “The ligands were prepared as described in [35] even..”. Correct the reference number in uppercase.
Done
- Page 12, “focusing the search within the spheres previously defined [Error!Bookmark not defined.]..” Missing reference.
The reference was added.
- Page 13, “..The predictive power of the final consensus models was assessed by subdividing the dataset in training (70%) and test (30%) sets and by repeating this validation 5 times to minimize the randomness..”. Stating the criterion upon which the dataset was subdivided (random?) would be beneficial.
Modified as suggested by adding “randomly”.
- Page 15, correct the formatting of Reference 20 (no uppercase)
Done